# Carbon Footprint as a Lever for Sustained Competitive Strategy in Developing a Smart Oenology: Evidence from an Exploratory Study in Italy

**Luigi Galletto and Luigino Barisan \***

Interdepartmental Research Center for Viticulture and Enology, Via XXVIII Aprile 14, 31015 Conegliano, Italy; luigi.galletto@unipd.it
\* Correspondence: luigino.barisan@unipd.it; Tel.: +39-0438-450-475

**Abstract:** In the wine sector, the choice of a sustainable strategy based on smart marketing has gained more relevance due to the growing importance of sustainability. The literature illustrates a multiplicity of perspectives, wherever firms are committed to improving sustainability and market performance. This exploratory paper aims to make sense of the extant literature by analysing 10 case studies in Italy, focusing on sustainable competitive resources and strategies, considering the role of CF (Carbon Footprint) as a crucial factor. The research has considered a complementary theoretical framework based on both Resource-Based Theory and Competitive Advantage Theory. Data were analysed by descriptive statistical techniques. The results show a bundle of unique resources and strategies in pursuing firm performance, wherever CF may lead to significant sustained competitive advantages in firms' value capture (ex. image and reputation and customers' relationships loyalty, entrance into new foreign markets). Findings highlight that perceiving the costs and benefits of investments in lowering CF may guide to a more accurate understanding of the value-creating from the different type of eco-innovation for building tailor-made communicational and marketing strategies.

**Keywords:** winemaking; value capture mechanism; physical capital resources; emission reduction practices; eco-innovation; environmental impact

---

## 1. Introduction

In the wine industry, firms could opt to differentiate both their images and products based on diverse strategies in order to gain competitive advantages [1–3]. The success of differentiation strategies mainly depends on the perceived consumers' values and lack of imitation by the competitors. Therefore, firms usually refer to the implementation of multiples and strength sources in supply differentiation.

Among all environmental sources of supply differentiation, the adoption of a carbon footprint (CF) strategy (CFS) is gaining a role of primary importance within the wine sector as it involves leaders who play a vital role in ecosystems preservation and climate change mitigation [4–8].

From a global point of view, the Greenhouse Gases (GHG) Accounting Protocol is a CFS designed by the International Organization of Vine and Wine (OIV) for the wine sector. Its general purpose is to provide organisations, businesses and other stakeholders with a consistent method for identifying areas of emission reductions associated with vine and wine firms activities [9].

Following a process based on the enhancing of environmental performances, CFS' key drivers spring from different sources around the world put together from global, national and regional experiences in the firms and product manufacturing [10].

To start with, one of the most relevant drivers cited in literature appears when considering the CFS' model adopted by the most important European countries and that of the so-called New World producing countries. Among the latter, the United States, Australia, South Africa and New Zealand have implemented the International Wine Carbon Protocol (IWCP) in order to enhance wine eco-profiles, wine companies' sustainability and people's welfare [6,11].

As an example, Champagne was the first wine-growing region in the world to conduct a carbon footprint survey. It has reached −15% per bottle of GHGs reduction between 2003–2013 [4]. In addition, by adopting 2014–2020 CAP Rural Development Program, Champagne-Ardenne and Castilla La Mancha have planned to invest funds on the environment in order to support a shift towards low-carbon and climate resilient wine economy [12,13].

Adapted from International Wine Carbon Calculator (IWCC), Ita.Ca® represents the first Italian sector's strategy based on ISO 14064 and ISO 14067 standards that specify principles and requirements at the organisation and product levels respectively [14].

Moreover, internationally, a certain number of firms have voluntarily adopted CFS in order to search points of competitive advantage and to face climate change challenges first by reducing carbon dioxide emissions and, secondly, by increasing carbon dioxide capture and sequestration [15]. In New Zealand, Tree Hill developed a CFS, thus claiming to be the first global Carbon Zero certified winery [3]. Since 2007, the biggest Chilean wine industry is carbon footprint certified, committed to its value as a long-term strategy's management tool [16]. In 2006, Fetzer Vineyards, one of the top US wine firms, joined voluntary GHG reporting initiatives, based on California Climate Action Registry (CCAR). It has been promoting its image as an Earth-friendly winery, by using vineyards practices for storing (sequestering) carbon [17].

Finally, a CFS has been analysed on different kinds of wines (reds, whites, sparkling and organics), showing differentiated environment impacts that could be used as an eco-labelling strategy [18–21]. In this way, a *sine qua non* condition is the willingness by consumers to pay the price higher than the cost of adoption of those environmentally-friendly vitivinicultural practices for a sustainable wine [22,23]. Indeed, wine product's environmental sustainability represents an element that has recently been gaining more and more importance in consumers' buying choices [24,25].

A larger number of firms are beginning to take into account factors related to wine production's environmental sustainability. This fact is in line with a growing demand for products with environmental certification in the years ahead [6,26]. Therefore, a more specific approach based on an environmental management system (EMS) may result more effective in developing a firm competitive advantage and in improving positioning on the market [27–30].

In this ambit, over the last decade, the CF has shown a growing body of research arising from different perspectives [6,31–33]. Nevertheless, no study, at least in Italy, has analyzed sustainable strategies at firm and product level, considering the role of CF as a key factor. A review of published material conducted for the wine sector indicates that there is limited empirical knowledge on how firms integrate CF into companies' supply strategy.

This explorative paper addresses four related issues assuming carbon footprint as a crucial factor for pursuing firms sustained competitive advantages [34–36] by:

- Exploring bundle of resources deployed by firms to improve their ecological carbon footprint at the whole production cycle (grape production, winemaking and bottling, and commercialisation).
- Linking bundle of resources to strategies conceived and adopted by firms aimed at improving their efficiency (i.e., improving cost reduction) and effectiveness (i.e., improving use value and value capture), by considering firms' manager opinions and perceptions.
- Focusing the attention inside the firm to core activities that create use value and subsequently, contribute to realising new exchange value for customers.
- Measuring the expectations of firms' managers about impacts for having implemented a CFS, aimed at improving capture value and its relationships with customers.

The paper is structured as follows. Firstly, the theoretical framework and the methodology used to explore CFS is introduced. Secondly, the major findings from data analysis are presented. Finally, the concluding discussion of the work is reported.

## 2. Conceptual Framework

The study starts from a theoretical framework that, when understanding sources of firms competitive advantages, considers two models: the Resource-Based Theory (RBT) and Porter's Theory of competitive advantages (CAT) [35] as they practice complementary (dual) roles [34,37–42] in implementing the firm's strategy. The theoretical complements between Porter's conceptual structure and the RBT's have been summarised by Foss' contribution [43]. This theoretical framework begins with the resources that the firm possesses, it considers their potential value, and it defines the strategy that allows the achieving of the maximum value in a sustainable way [44].

In this sense, according to the RBT, it is assumed that the firm is heterogeneous in the resources it uses as sources when implementing a strategy of sustained competitive advantage [34,45,46]. According to Barney [34], the firm achieves a sustained competitive advantage not only when it is able to implement a value creation strategy that is not simultaneously imitated by other current and potential competitors but also when competing companies are incapable of obtaining the same benefits related to the considered strategy.

The firm's resources are conceived and selected within the firm among those that allow seizing external opportunities better, trying to maximise the firm's performance [47,48]. The attributes used in the value-creating strategies can be classified, by convention, in three macro-categories: physical capital resources [49]; human capital resources [50] and organisational capital resources [51]. The former concern the technologies used in the firm, such as plants and equipment, access to raw materials, and so on, while the latter concern training, acquired experience, business and technical intelligence, personal creativeness, relationships, etc. The last defines activities such as task allocation, coordination and supervision, as well as relations within its environment.

In the view of the value chain introduced by Porter [52], one takes into account the firms' resources to isolate potential sources that underlie the firms' competitive advantage. In the RBT conceptual approach, one takes into account a further logical step compared to Porter, analysing the resources owned by the firm, heterogeneous and, to some extent immobile, which have the attributes to be sources of sustained competitive advantage [34]. In order to have this potential, the firm's resources must be characterized by four attributes: valuable, exploiting opportunities and/or neutralizes threats; rare (unique or a bundle of resources) in the sense that they have to compete with current and potential competitors; imperfectly imitable; and which do not have strategically equivalent substitutes that are of value, rare and imperfectly imitable and from which the same strategy can be implemented effectively and efficiently as in the original resource.

## 3. Carbon Footprint Strategies as a Lever of Sustained Competitive Advantage in the Italian Wine Sector

According to some RBT authors, the resources' attributes that most likely represent superior performance factors for firms must be held by them [53] and having the features of "isolating mechanisms" [46] or constitute "bundles of unique resources".

Within this perspective of strategic business management (RBV theory), the firm's choice to adopt the carbon footprint certifications (organization or product) can represent to firm's strategy of how it can outperform others in the market where it operates; this strategy uses a bundle of unique resources (i.e., capital physical resources and capabilities resources) to achieve effectively both objectives, improving environmental performance and improving economic performance [54].

For the firm, the CF could be as a lever of sustained competitive advantages, concentrating the sustainability factors' forces used in competition at firm and product levels [34,38,55–59]. Therefore, in Italy, the CFS in the wine sector has been developed starting from 2009, through a complex

social commitment, by a limited number of firms (rare efforts) stimulated to undertake virtuous and sustainable paths, aimed at preserving the environment by adopting lower impact on vineyard practices, such as grassing and the use of organic fertilizers, the reduction of emissions from non-renewable energy sources and the optimization of production processes of the wine making (causal ambiguity and socially complex links) [14,60]. These characteristics, combined with a complex approach to firms' implementation, make these competitive advantage resources not easily transferable to other companies (non-substitutable) [14,60].

Nowadays, in Italy, the environmental sustainability strategy that takes into account CFS relies mostly on two institutions: VIVA, founded in 2011 by the Italian Ministry of the Environment and Equalitas, stemmed from Federdoc and Unione Italiana Vini, which was established in 2015. They provide overarching principles in differentiated directions for pursuing sustained competitive advantages throughout their codes [61,62], which can be seen as an inside-out process [44].

Currently, the firms with a CF certification amount to only 14 firms and 56 wines own a product carbon footprint (Appendix A Table A1). The majority of those involved in adopting business CF is located in the central (50%) and southern regions of the country (28.6%) and only a minor share in the northern regions (21.4%), thus constituting pioneering brands in the Italian wine sector with competitive advantages over new entrants.

Equalitas (64% of firms) and VIVA (36%) have calculated firm and product CF differently (Appendix A Table A1). Equalitas has conceived the "functional unit" taken as a reference to report the total firm emissions data to a unitary production element (production phases); in the vineyard, it considers the kilogram of grapes, in the cellar the litre of bulk wine and for bottling phase the standard bottle (0.75 L), within VIVA's wine sustainable program, the life cycle of wine bottles includes four major phases: vineyard management, the transformation of grapes into wine and bottling, distribution of bottles, refrigeration and disposal (called respectively vineyard, firm, distribution and consumption). The total emissions are divided into three categories: direct emissions generated from resources of ownership or under the control of the firm, where ambit 1 affects direct emissions; indirect emissions due to the firm's energy consumption, in which ambit 2 refers to indirect emissions from energy consumption; indirect emissions generated from resources not owned or controlled by the firm that are not due to energy consumption and that are referred to ambit 3 regarding, other indirect emissions.

Firms with VIVA and Equalitas' sustainability certifications showed a wide variability of carbon footprint magnitude along winemaking' phases and ambits; the reported wide differences in firm environmental performance within the groups suggesting that best practices and performance are linked so that a firm fits into the class with higher quality standards [63–69]. The firm that owns these resources contributes to generate a better reputation and image over time [70] which, in the case of assets that affect the reduction of the CF, are not readily transferable in all organisational contexts.

Finally, in close line with Peteraf and Barney [37], this resources may represent a competitive advantage as a superior differentiation or lower costs by comparison with the marginal (break-even) competitor in the wine market. Considering insights from these viewpoints may lead to more effective marketing firm sustained strategies.

## 4. Materials and Methods

### 4.1. Research Approach and Design

From a theoretical point of view, the research work takes first into consideration that a differentiation strategy is a multidimensional approach [28,71]. Indeed, given a differentiation based on information on product features and the optimal mix of these, a sustained competitive advantage can be determined because consumers tend to be loyal to the pioneering brands over new entrants in the market [72]. This assumption was tested by Schmalensee [73] and confirmed for the wine market, which is characterised by the high number of brands and imperfect consumer information on wine

quality [74–76]. Therefore, it becomes harder for later entrants to convince consumers to learn about wine credence qualities than it was for pioneering brands.

Attempting to move beyond previous research [3] by considering a more focused source of sustained competitive advantages for the Italian wine sector, a case study [77] was carried out using CF as the crucial factor of an effective strategy of environmental sustainability.

The study methodology was developed in four stages [10,30,60,78–81]. Firstly, a literature review was carried out to identify key concepts and variables describing firm resources and strategies involved in sustained competitive advantages from sustainability and market performance viewpoints. The purpose was to extract the key variables to be used in the second stage of the research.

Secondly, qualitative data from some Italian firms that already dealt with the sustainability issue and official statistics in order to explore the degree of involvement of Italian wineries in environmental sustainability were gathered.

In the third step, considering variables emerging from the literature review and qualitative research, a survey questionnaire (see the Supplementary) was implemented. The survey questionnaire has allowed building a richer array of evidence. The research combines quantitative and qualitative techniques, approaches and concepts. The research constructs were measured on the knowledge of the firm representative's perception on CF-related investment costs and benefits and firm practices. Gathered data on CF follows the VIVA and Equalitas procedures, where Ita.Ca is the reference tool in calculating CF, which distinguishes between Enterprise Protocol and Product Protocol [60].

In the fourth step, at the end of the questionnaire-development, the pretesting was made in order to ascertain how well it works. According to Hunt and et al. 1982 five issues involved in the pretesting questionnaire were taken into account [82]. Firstly, the order and statements of the questions; the choice and relevance of the items, the accuracy of the interpretation of the answers and any respondents' difficulties; linking of the questions with the research objectives. Secondly, personal interview pretesting through in-depth face-to-face interviews with two kinds of interviewers was chosen. Thirdly, the study has considered interviewers with different degree of competence such as marketing managers from an Italian Carbon footprint certified sparkling wine firm located in the Veneto Region and University of Padova's Academics. Academics were two Professors (tenured positions) in viticulture, oenology and wine marketing. Fourthly, therefore the firm selected was similar to the target of respondents, following the ISO/TR 14069 certification standards since 2014. Fifthly, given the explorative nature of the study, only one pretest on firms was made considering the limited target population, but it was integrated with Academics viewpoints for giving a view that goes beyond a merely checking of questions. Finally, thanks also to their valuable contributions and final adjustments, the questionnaire has been submitted personally to firm representatives.

According to Bowman and Ambrosini [36,83,84], the study considers (Sections 4–6) the term use values (UVs) as the properties of products and services that provide utility to the firms. Therefore, along with the vine and wine supply chain, the properties of firms' inputs are considered separable. On the one side, they would include any bought-in physical capital resources (ex. plants, types of machinery, equipment, etc.), which are UVs in the form of owned by the firms. On the other side, they include human (capital) inputs, which are UVs in the form of performed activities and strategies. Human capital's activities about the procurement of inputs (ex. physical capital resources and practices adopted) into the firms' supply chain are based on beliefs about their usefulness in the UV creation process by pursuing a CFS. The sources of new UVs is attributed to firms' profits derived from human capital that perform heterogeneously across firms for their different capabilities. Firms' UVs are converted into exchange value (EVs) when they are sold in wine markets (i.e., realised at the point of sale). Firm's value capture (VC) for having conceived and implemented a CFS depends on their bargaining power which is determined by their perceived power relationships with customers.

Within this methodological framework, the study has explored three Resource-Based-View strategies linked to CFS' mechanism across the vine and wine supply chain's phases. More in detail, the CFSes implemented by firms and their economic and market impacts were evaluated,

by considering their: (a) cost advantage, where firms performed below their average cost (C-) and above average their consumer surplus (CS+); (b) premium price advantage (M+), where firms perceived user value is higher (PUV+) than their previous PUV (i.e., before adoption of the CFSes), enabling the firm to charge premium prices (P+) due to the fact that customers experienced superior firms and/or wines' performance (CS+) lead to superior profit; (c) no premium price is charged, but superior perceived user value (PUV++) would attract more customers; even if the average costs remained unchanged the superior sales volumes and/or experience-based resources and firms practices could lead to superior profit.

In attempting to better deal with the four fundamental issues previously defined in the study aims linkages with the three Resource-Based-View strategies were carried out. In order to consider CF mechanism the questionnaire was conceptualized by the followings assumptions [36,83,84]:

- Differences across the firm's strategies are attributable to the particular amount of bundle of resources deployed and practices adopted over the time.
- New UVs creation was submitted to firms' subjective judgements on the acquirement of inputs into the whole production processes. For this reason, on the one hand, firms were asked if they are performing lower than previous average costs (C−) by having adopted a CFS; on the other hand, firms' managers were asked about the perceived usefulness of the bundle of resources deployed and practices adopted (i.e., PUV+ and PUV++) by having adopted a CFS.
- Within this UVs creation process intervention of firms' human capital was assumed as necessary for implementing CFS (i.e., through core communicational activities).
- The realisation of VC by firms, regarding the power of relationships between firms and customers, was explored by considering some crucial indicators (i.e., image and reputation improvements, customers' relationships loyalty improvements, entrance into new domestic distribution channels, entrance into new foreign markets and distribution channels) and their relationships.

*4.2. The Semi-Structured Questionnaire Used*

The semi-structured questionnaire was framed within five parts as follows. The first part of the questionnaire includes firm general information, structural and production data. The questions concern legal structure, employment by job task, vineyard surface by type of production, firm size by class of turnover, total cellar capacity and certifications (ex. ISO/TR 14069, ISO/TS 14067; Emas, Magis, organic, biodynamic, VIVA, Equalitas and others).

The second part deal with the following market data: total bottles sold, wines' share market on the Italian market and by its distribution channels (i.e., direct sales, large-scale retail distribution, wholesales and others channels). According to the Global Trade Atlas (GTA)'s trade database, the followings exports markets of Italian firms were identified among the most relevant Countries: United States, United Kingdom, Germany, Russian Federation, Switzerland, China, others European Countries (France, Spain, Denmark, Finland, Iceland, Norway, Sweden, Estonia, Latvia, Lithuania, Poland, Czech Republic, Slovakia Bulgaria, Hungary, Romania, Slovenia, Greece, Ukraine) and others Asian Countries (Japan and India) and rest of the world Countries (Australasia, Africa, others Americans and Asian).

The third part considers key concepts on CF-related resources and their perceptions from firm managers' viewpoints. They deal with both the interviewers' view on some related aspects, as motivations on the adoption of a CFS, impact in term of improving average costs reduction and usefulness of resources deployed and practices adopted, interpreted in the light of sustained competitive advantages approach. CF data were collected at both firm and product levels (for whites, reds and sparkling wines) and are expressed in term of $CO_2$-equivalent. In this part, it was evaluated the impacts of the CFS on both reducing the overall costs and increasing firms' UVs. Hence, this part relates to the physical capital resources employed in the followings production phases: vineyard, winemaking, sales and logistics. Benefits include the evaluation of the physical capital resources used in CF reduction. Firm practices are referred to procedures followed in the vineyard, or way of

wine-making and along the logistic and sale processes (i.e., use of lightened glass for wine bottles, 'green' packaging, optimisation of freight vehicle, etc.).

The fourth part deals with the evaluation of the impacts of physical capital investments and practices adopted on reducing their average costs. Potential competitive advantages concerning premium pricing advantages were assessed too.

The fifth part deals with some VC key elements considered in the study and communicational tools aimed at seizing potential opportunities arising from the market through the CFS. These aspects reflected some elements identified by Lash et al. (2007) [29]. For each question, the firm representatives were asked to express a judgment on economic benefits of a CFS on improving image and reputation, and improving customer's relationships loyalty, entering in new channels in domestic and foreign markets. These questions are fulfilled by agreeing on communicational tool strategies that can be more suitable for the development of a CFS.

Parts ranging from 3 to 5 include 7-points Likert scale questions ranging from 1 ('not at all important or useful') to 7 ('extremely important or useful'). These questions combined with quantitative ones are based on perceptions, opinions, motivations and expectations of firms' representatives about the resources deployed, the practices adopted and the strategies implemented for improving the CF along the whole supply chain and by major production phases [11,79,80].

The whole questionnaire, dataset and data description are available in the annexed materials.

### 4.3. Firm Sample and Data Analysis

The firm's sample has been selected on purpose by identifying ten Italian firms, representing about 20% of all the firms listed in the Ita.Ca's protocol and 71% of the major Italian sustainability certified programs (VIVA and Equalitas). In more detail, firms were chosen among those who adopted ISO/TR 14069 standards (80% of cases) or ISO/TS 14067 standards (60%). Particular attention was paid to guarantee a representative variety of carbon footprint standards across wine-producing areas, firms' sizes and wines portfolio. Their locations were chosen in different wine regions in Italy (Veneto, Friuli Venezia Giulia, Lombardy, Tuscany, Umbria, Lazio, Campania and Sicily) (Figure 1) [60]. The study sample has considered three different firms 'size on the basis of the total number of bottles sold annually (standard equal to 0.75 L): 30% of relatively small units (less than 150,000 bottles); 20% of medium-sized firms (from 150,000 to 500,000 bottles) and 50% of large firms (over 500,000 bottles). Small firms are underrepresented in the study sample; indeed, the larger a firm, the more likely it is to be ranked by having carbon footprint standards.

Firms were previously contacted by phone and subsequently appointments were made with the owners in order to submit face-to-face the questionnaire. The preliminary phase of the research was particularly challenging, because of the required quantification, together with the interviewees, of the exact costs and benefits of both investments and firm low-environmental practices aimed at reducing CF.

Given the number of investigated wineries, a case-by-case analysis was not performed. Rather, a sample analysis was undertaken, by using basic descriptive statistics.

Sample data analysis has been performed by using basic descriptive statistics. Mean and standard deviation were employed for parametric variables, while median and interquartile difference were employed for Likert scores, considering each score value as a single class. Although some studies manage them as numerical values, the lack of normal distribution for most of them has implied this choice.

The Spearman's rank correlation coefficient has been employed to verify possible linkages among Likert scale variables, CF-related variables and other aspects of the sample firms. Although these coefficients are obtained by a rather small sample, they represent 20% of the population of the pioneering firms that have undertaken the CFS path and, therefore, rather than supporting any linkage of causality, they can reveal at least significant associations among variables, for a better understanding of the CFSes in the Italian wine industry.

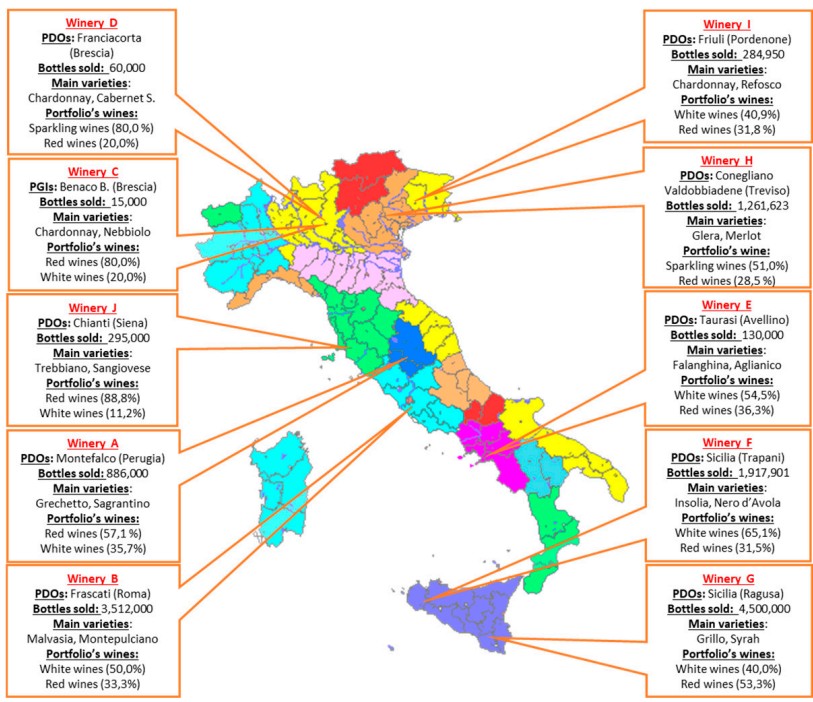

**Figure 1.** Location of the firms selected in Italy: 10 case studies.

The results analysis of results is reported into five sections: first, an introduction to selective descriptive results of their structure and market are presented; second, investments on specific physical capital resources and firm practices aimed at GHG emission reduction were evaluated both in terms of amount of the investment values and their specific impacts on cost reduction; third, the impacts of CFS-related investments as a whole and for vineyard, winemaking practices, logistics and sales were evaluated [85,86]; fourth, the impacts of CFS-related investments, measured considering some economic and market aspects were estimated; finally, taking cue from linkages between RBT and CAT, the firm benefits in performing CFS were analyzed considering their relation with image and reputation, market performance, consumer loyalty along communication strategies.

The results were subsequently validated by a focus group of Academics, firm managers and technical managers of the certification bodies.

## 5. Results

### 5.1. Structural and Market Data

The vineyard of firms' sample ranges from a minimum of seven hectares to a maximum of 262 hectares. The produced bottles are on average about 1.3 million bottles. White wines prevail (56.8% of the total bottles sold), followed by red and rosé wines (37.7%), while sparkling wines make up the rest (5.5%). The economic results of the firms by turnover class, confirms a wide variability in the distribution of sales value, showing in the majority of cases relatively medium-low revenues (70% of firms), below 5 million euros, while a smaller number of firms towering above 5 million euros (30%). Firms' core competencies analysed by phases of the production cycle showed a quite balanced number of employed across major phases of the supply chain (from grape production to wine production, to marketing and sales). It is worth noticing that all the firms own at least one environmentally-related certification; organic viticulture rely on 40% of firms, and organic production has a relatively high share on their total wine production (26.5%). The firms of the sample pursued a multi-channel strategy and oriented towards various export markets. The firm sample's profile has been summarised in Appendix A Table A2, while the complete structural and market data are reported in the annexed materials.

## 5.2. Investments on Physical Capital Resources and Firm Practices for GHG Emission Reduction

To explore resources owned by the firms towards a lower GHG emission of the firms, the first part of questions was elicited from a resource-based perspective. The analysis of physical resources deployed by firms in the winemaking phase shows heterogeneous endowment concerning both renewable energy systems and energy efficiency equipment, which can be summarised in Table 1. Considering results from grape production phase no significant CF-related resources were found.

**Table 1.** Bundle of physical capital resources deployed by firms, 2009–2016.

| Physical Assets | Frequency | Period (Years) | Mean Value (€) |
|---|---|---|---|
| **Renewable energy systems:** | | | |
| **Photovoltaic** | 80% | 2002–2016 | 413,750 |
| **Thermal solar** | 50% | 2009–2014 | 13,000 |
| **Biomass** | 30% | 2011–2015 | 1,320,000 |
| **Hydroelectric** | 10% | 2012 | 200,000 |
| **Energy efficiency equipment:** | | | |
| **Tank insulation** | 30% | 2011–2013 | 8667 |
| **Natural ventilation** | 30% | 2011–2013 | N.A. |
| **Natural light usage** | 30% | 2011–2013 | N.A. |
| **LED light usage** | 40% | 2011–2015 | 3640 |

Source: authors' elaboration.

As far as renewable energy systems are concerned, photovoltaic plants are the most widespread (80% of cases) when only one competing firm possesses a hydroelectric plant, the relatively scarcest resources deployed; it is followed by 30% of firms that acquired biomass plants, averagely the most costly-to-imitate resources. A variety of equipment was held by few firms for energy efficiency improvements. Some of them are firm-specific resources that determine impacts on reducing energy uses (i.e., natural ventilation and natural light usage).

Considering results from logistics and sales phases, it emerged that only one firm acquired (in 2014) a more environmentally-friendly transport such as hybrid vehicle. Among that firms practices that tend to reduce GHG emissions quite frequent (60% of cases) are the delivery optimization. This is based on the adoption of improvements of transportation models (e.g., optimization of freight vehicle utilization ratios and transport routes, fixed delivery days, etc.) aimed at reducing both $CO_2$ emissions and average cost as part of their logistics design [85,86]. The composition of firms' portfolio of investments in physical assets was found over a relatively more extended period for renewable energy systems investment, rather than for energy efficiency equipment.

Among the environmental performance, some wineries can produce a standard bottle of wine emitting only 1.22 kg of $CO_2$, rather below the average recorded for a bottle of wine (approximately 2.2 kg $CO_2$/bottle).

## 5.3. CFS and Economic and Market Impacts

Table 2 shows the impacts of firms' physical assets and practices on improving costs reduction by major phases of the production chain and as a whole. Managers' opinion on a bundle of winemaking practices and sales and logistics were rated as important to lower their relative average costs (median score equal to 5) and rated moderately importance for vineyard practices. Moreover, the impacts of firm overall physical capital resource on cost-reducing (even considering all logistics and sales costs) were assessed as important.

Results on specific market factors deployed by firms in minimising GHG emissions evidence synergies in improving their business effectiveness. This is attributable to the particular bundle of macro resources used (i.e., physical capital and energy efficiency investments) for lowering their winemaking's average total costs. Natural light usage and thermal solar jointly with biomass plants, tank insulation and LED light usage are perceived by firms 'manager among the most cost-effective,

respectively ranging from a median score of extremely important for the first two and rated as very important (median scores equal to 6) for the others (Table 3). It is also worth noting that the interquartile range of scores showed a relatively low measure of spread across firms resources, ranging from a minimum of 1 to a maximum of 2 points.

**Table 2.** Median and interquartile range scores [1] of managers' opinions on physical capital resources' economic impacts on improving costs reduction by vineyard, winemaking, sales and logistics sustainable practices and overall physical capital investments.

| Physical Capital Resources and Practices | Median | Interquartile Range |
|:---|:---:|:---:|
| Firm overall physical capital investments | 5 | 2 |
| Firm practices in the vineyard | 4 | 1 |
| Firm practices in winemaking | 5 | 2 |
| Firm practices in sales and logistics | 5 | 0 |

Source: authors' elaboration. [1] Notes: rating score is a 7-point Likert scale (ranging from 1 = not at all important to 7 = extremely important).

**Table 3.** Median and interquartile range scores [1] of managers' opinions on the bundle of physical capital resources impacts on improving cost reduction.

| Physical Capital Resources | Median | Interquartile Range |
|:---|:---:|:---:|
| **Plant investments:** | | |
| Photovoltaic | 6 | 2 |
| Thermal solar | 7 | 1 |
| Biomass | 6 | 1 |
| Hydroelectric | 4 | - |
| **Energy efficiency investments:** | | |
| Tank insulation | 6 | 2 |
| Natural ventilation | 6 | 1 |
| Natural light usage | 7 | 1 |
| LED light usage | 6 | 2 |

Source: authors' elaboration. [1] Notes: rating score is a 7-point Likert scale (ranging from 1 = not at all important to 7 = extremely important).

At this stage, it seems reasonable to consider the link between the resources deployed by firms and the CFS they should pursue. Results from the Spearman rank-order correlation tests showed correlation coefficients ranging between 0.64 for the impacts on reducing winemaking costs ($p$-value < 0.05) and 0.57 for overall firm costs reduction impacts ($p$-value < 0.1) (Table 4). No significant impacts were found between the usefulness of CF measurement and GHG in the reduction of logistics and sales costs.

**Table 4.** Spearman rank-order correlation ($r_s$) [1] coefficients between the usefulness of CFS' impacts on cost reduction by firms' whole production cycle and their major phases.

| Variables | $r_s$ |
|:---|:---:|
| On overall firm costs reduction | 0.566 (0.088) * |
| On winemaking costs reduction | 0.640 (0.046) ** |
| On logistic and sales costs reduction | −0.261 (0.617) |

[1] Notes: rating score is a 7-point Likert scale (ranging from 1 = not at all important to 7 = extremely important). * Significant at the 10% level, ** significant at the 5% level.

In answering a question on the need of a price increase following the choice of a low environmental impact production based on CFS, only two firms considered adequate around +10% price increase per bottle sold below 5 € per bottle (i.e., basic and popular premium wines segments), a price increase in the range between 10% and 20% per bottle priced between 5 and 10 € (premium and super-premium wines) per bottle and at least 20% per bottle increase per bottle priced above 10 € (super premium

and ultra-premium wines). The other firms indicated no need of a price increase strategies due to the adoption of a CFS, considering already significant the advantages in term of the achievement of other firm objectives (see sub-par. below).

*5.4. CFS and Firms' Core Communications Activities*

The analysis goes further by focusing on how the bundle of firms resources should be deployed and combined, through the actions of human capital, by considering the usefulness of CFS as a driver in firms' communication (Table 5). Results showed that fairs, wine tasting events, advertising by specialized press and communication at the firm's shop are all reasonably perceived by firm managers with median scores between very important and important; the evaluations are more concentrated for fairs (IQR = 1) and more spread for others communicational tools (IQR range from 3 to 6), likely reflecting previous experiences towards each tool.

**Table 5.** Median and interquartile range scores [1] of perceived usefulness of CFS as key factor in the communication of wines with low environmental impacts.

| Communicational Tools | Median [1] | Interquartile Range |
|:---:|:---:|:---:|
| Fairs | 5 | 1 |
| Wine tasting events | 6 | 5 |
| General advertising | 4 | 6 |
| Advertising by specialised press | 6 | 4 |
| Online campaign | 5 | 4 |
| Sponsoring | 4 | 3 |
| Firm point of sale communication | 6 | 4 |
| Television | 3 | 4 |

Source: authors' elaboration. [1] Note: rating score is a 7-point Likert scale (ranging from 1 = not at all useful to 7 = extremely useful).

Moreover, results pay attention to the firms' CFS to core communicational activities that are more usefully associated on GHG emission reduction, creating use value and subsequently, contributing to realising exchange value for customers (Table 6).

**Table 6.** Spearman rank-order correlation ($r_s$) [1] coefficients between firms' communicational tools and CFS' key concepts of the impacts in improving GHG emissions reduction and their usefulness in value creation process.

| Communicational Tools | Firms' PCR Deployed for Improving GHG Reduction ($r_s$) | Firms' WM Practices for Improving GHG Reduction ($r_s$) | Improving Firms' Image and Reputation ($r_s$) | Improving Customers' Relationship Loyalty ($r_s$) |
|:---:|:---:|:---:|:---:|:---:|
| Fairs | 0.677 (0.032) ** | 0.898 (0.000) *** | 0.538 (0.109) | 0.801 (0.005) *** |
| Wine tasting events | 0.733 (0.016) ** | 0.771 (0.009) *** | 0.374 (0.287) | 0.851 (0.002) *** |
| General advertising | 0.505 (0.136) | 0.926 (0.000) *** | 0.121 (0.740) | 0.598 (0.068) * |
| Specialized press' advertising | 0.159 (0.660) | 0.479 (0.161) | 0.464 (0.176) | 0.323 (0.363) |
| On line campaign | 0.662 (0.037) ** | 0.742 (0.014) ** | 0.193 (0.593) | 0.767 (0.01) *** |
| Sponsoring | 0.549 (0.099) * | 0.780 (0.008) *** | 0.247 (0.491) | 0.770 (0.009) *** |
| Cellar sale's communication | 0.664 (0.036) ** | 0.431 (0.214) | 0.294 (0.410) | 0.735 (0.015) ** |
| Television | 0.481 (0.160) | 0.936 (0.000) *** | 0.120 (0.741) | 0.691 (0.027) ** |

[1] Note: rating score is a 7-point Likert scale (ranging from 1 = not at all important/useful to 7 = extremely important/useful). PCR: physical capital resources. WM: winemaking. * Significant at the 10% level, ** significant at the 5% level, *** significant at the 1% level.

The rank-order correlation coefficients toward improving firms' communication on GHG emission reduction for the physical capital resources deployed range around 0.70 for wine tasting, fairs, cellar sale's communication and online campaign (*p*-value < 0.05). In addition, firms practices show

correlation coefficients towering above 0.90 for some categories like general advertising and television (*p*-value < 0.01). It is also worth noting that the magnitude of the association between the improvements of customers' relationship loyalty scores and all communicational tools is quite high except specialised press' advertising. In particular, the Spearman rank-order correlation coefficients are relatively high for fairs and wine tasting events reaching values above 0.80.

*5.5. CFS and Value Capture*

As far as expectations of firms' managers about impacts for having implemented a CFS are concerned, some key concepts were associated with firms' value capture.

Firstly, an improvement to the firm's image and reputation was rated as one of the most important goals towards a low environmental impact wine production, when considering CFS' usefulness as crucial factor. The same median score relates to the entrance into new foreign markets and channels, although less convergence among the respondents (Table 7). The interquartile range shows a high degree of agreement among respondents. This is followed by customers' relationships loyalty and entrance into new foreign markets and channels that ranks second being rated as important (both with median scores equal to 6).

**Table 7.** Median and interquartile range scores [1] of perceived usefulness of CFS based on the bundle of physical capital resources in the value capture process.

| Objectives | Median [1] | Interquartile Range |
|---|---|---|
| Entrance into new channels in the domestic markets | 5 | 5 |
| Entrance into new foreign markets and channels | 6 | 4 |
| Firm image and reputation improvements | 7 | 1 |
| Customers' relationships loyalty improvements | 6 | 2 |

Source: author's elaboration. [1] Notes: rating score is a 7-point Likert scale (ranging from 1 = not at all useful to 7 = extremely useful).

Secondly, attention was paid to the relationships between buyers and sellers, considering CFS' usefulness in firms' value capture process. Nevertheless, when disaggregated to the subcategory of value capture, some categories show weak associations with CFS (i.e., entrance into new foreign markets and channels, improving firms' image and reputation). On the contrary, it is worth noting significant correlations between CFS and customer's relationships loyalty both for GHG emissions reduction ($r_s = 0.80$ and *p*-value < 0.05) and for use value creation process ($r_s = 0.65$) (Table 8).

**Table 8.** Spearman rank-order correlation ($r_s$) [1] coefficients between CFS based on cost reduction, CFS usefulness in the value capture process.

| Variables | CFS' Firms Cost Reduction ($r_s$) | CFS' Firms Usefulness for Improving Business Performance ($r_s$) |
|---|---|---|
| Entrance into new channels in the domestic markets | 0.590 (0.073) * | 0.446 (0.200) |
| Entrance into new foreign markets and channels | 0.479 (0.161) | 0.387 (0.269) |
| Improving firms' image and reputation | 0.4376 (0.206) | 0.502 (0.139) |
| Improving customers' relationship loyalty | 0.804 (0.005) ** | 0.650 (0.042) ** |

[1] Notes: rating score is a 7-point Likert scale (ranging from 1 = not at all important/useful to 7 = extremely important/useful). * Significant at the 10% level, ** significant at the 5% level.

To conclude, Table 9 shows Spearman rank-order correlations among firms' image and reputation and relationships with customers' loyalty ($r_s = 0.56$; *p*-value < 0.1), in which CFS serves as a critical mediating variable in the value capture process. A relatively high correlation between customer's loyalty and entrance into new foreign markets and channels was found ($r_s = 0.65$; *p*-value < 0.05). On the other side, relationships with customers' loyalty show no direct association with entrance into new channels in the domestic markets.

**Table 9.** Spearman rank-order correlation ($r_s$) [1] coefficients between improved customers' relationship loyalty for firms and key variable in the value capture process.

| Variables | Firms Improving Customers' Relationship Loyalty ($r_s$) |
|---|---|
| Improving firms' image and reputation | 0.556 (0.094) * |
| Entrance into new channels in the domestic markets | 0.547 (0.102) |
| Entrance into new foreign markets and channels | 0.648 (0.043) ** |

[1] Notes: rating score is a 7-point Likert scale (ranging from 1 = not at all useful to 7 = extremely useful). * Significant at the 10% level, ** significant at the 5% level.

## 6. Discussion

Looking at the previous outcomes, some points seem worth being addressed.

The findings of the research showed that firms own a bundle of physical capital resources and some of them are relatively scarce and costly-to-imitate. It is also shown that over time firms tend to acquire new assets into their bundle of resources and supply chain practices in order to protect them from imitation through isolating mechanisms. New investments relate almost entirely the winemaking phase. The good correlation between the usefulness of CFS and the impact in reduction in costs related to the winemaking process, seems to support this choice. The decrease in GHG emissions by means of physical assets in this phase is undoubtedly an easier way to achieve a better CF than in the grape production phase, where techniques are widely standardised. Innovations like biofuels are actually quite rare, and fixed automatic spraying plants or electric tractors are just at the prototype level of development. Undoubtedly, some firms were able to benefit from specific funds for investments in solar (both photovoltaic and thermal) energy, available since the first decade of this century (financed by Rural Development Plan), which made them the most adopted within the physical assets. This fact stresses the importance of the public subsidies in speeding up the transition towards a more sustainable production also in the wine industry and may explain why investments in renewable energy systems prevail on energy efficiency equipment.

The procurement of inputs into the wine production process is assessed as cost-reducing and respondents believed in the impacts of the resources deployed for GHG reduction on the use value creation process. In addition, the CF levels that the sample firms achieved confirm this fact. In summary, managers' opinions on overall physical capital investments and winemaking practices as a whole are both scored important in achieving substantial cost reduction. More in detail, analysis evidenced that specific winemaking assets possessed by firms (i.e., natural light usage and natural ventilation equipment, biomass and thermal solar plants) receive higher expectations for improving their efficiency than others. It is also worth stressing the link between CFS and improving reduction costs of the winemaking process (Table 4). Moreover, results show beneficial relationships between firms' overall costs reduction and usefulness of CFS. This fact appears in line with a price increase as consequence of the adoption of a CFS, which two firms declared, given that better pricing can be coherent with better positioning of the firm's wine portfolio.

The correlation coefficients between firms' CFS and core communicational activities evidence the tools that are more usefully associated with: improving cost reduction, creating use value and subsequently, contributing to realize exchange value between firms and their customers and value capture image and reputation and relationships of customer loyalty with customers [87]. Among these, wine tasting, participation in fairs and online campaign were assessed as major strategic tools in building CFS's profitable relationships for improving firm's performance (Table 6). This highlights the relevance of human capital competencies call for dynamic capabilities [88] in setting up and managing CFS, that have to consider the changes that shape the competitive environment. The fact that firms' image and reputation show no significant correlation with the importance of all communication tools does not imply that it has no association with them, rather it is likely to depend on the low level of variability that relates to this variable, which was reputed the most important CFS's objective by almost all the firms (Table 7).

Among the expectations about the usefulness of firms' CFS in the value capture process, customers' loyalty relationships improvements and entrance into new market and channels are rather important too. The former appears congruent with the sample firms' commitment toward sustained competitive strategy, while the latter seems the consequence of the general export orientation shown by most of them. As far as the results show the relationships among firm image and reputation and customer loyalty are concerned, it shows how firms CFS' value capture can serve as a critical mediating variable for attaining performance improvements.

Moreover, the significant association between the levels of importance of the consumer's loyalty improvement with the importance of the entrance into new foreign markets and channels appears to confirm the marketing advantage following the low environmental impact production choice, based on CFS. It might be argued that, by adopting a CFS, firms intend to pursue a long term competitive advange especially in the foreign markets, where a greater consumers' feeling to environmental issues than in Italy makes them easier to accrue the loyalty to CF-related brands.

## 7. Conclusions

Over the last years the world wine market has changed rapidly, and at these changes, firms have sometimes responded by putting into actions short-term strategies based on immediate mechanisms. However, this approach calls for the use of long-term business strategies [89] that should be based on more reasonable mechanisms to pursue sustained competitive advantages. Among innovative mechanisms used in firms' strategic management, the macro indicators of sustainability (carbon footprint, water footprint, biodiversity footprints) follow the mode of operation of a complex phenomenon through the combination of several factors [90]. Therefore, they can play a growing role beyond the goals of minimising the environmental impacts of supply chains' wine production as firms' levers for value creation.

In this context, the Italian wine sector is characterised by a variety of environmental approaches, where Equalitas and VIVA stand at the heart of the national sustainability program that has been undertaken by, among other things, heterogeneously conceiving and implementing CF as a reference tool in GHG emissions reduction [60–62].

This study aimed to explore on how a selected sample of Italian firms integrates CF into firms' marketing strategy, by considering CFS factors' forces used in competition at business unit (ISO/TR 14069) and product levels (ISO/TS 14067).

Although literature regarding CF is quite extensive and arising from different viewpoints, no study, at least in Italy has analysed firms strategies considering the theoretical framework and insights elicited from RBT and CA viewpoints. Likewise, there is not enough empirical evidence from wine producers perspectives.

Therefore shedding light on firms strategies based on the deployment of valuable and sustainable resources is particularly interesting for developing a supply that should overcome customers expectations. Indeed, CFS may represents an important asset for value capture in firms management by adopting effective innovations where human capital capabilities may play central roles through complex social commitments.

From the standpoint of the results concerning the theoretical framework adopted some implications of interest emerge. According to Bowman and Ambrosini [36,83,84], the study, despite its exploratory character, appears to provide internal agreement with RBT's theory derived from firms' specific bundle of resources and practices deployed by firms. On the other side, following additional CA's approach, it guides firms reasoning on external relations with customers for improving their value capture, and integration among them (i.e., through more suitable communicational tools, etc.).

Regards to the empirical evidence, the study showed how the firms of the analysed sample make the carbon footprint a suitable strategy of sustained competitive advantage. It derives from the benefits of the smart uses of the deployed resources in the supply chain and particularly in the winemaking phase. The firms' benefits were reported considering linkages across Resource-Based-View strategies.

A specific stock of valuable resources that was deployed by firms emerged. This has given efficiency and effectiveness along the winemaking process. In particular, the benefits relies on the fact that the firms have performed below their average relative costs and improved their perceived value (before the adoption of CFS). In this value creating the study has shed light as some firms' core activities are of relevance. Among these the wine communication tools, which strengthen the direct link with the customer, appear to be more effective in grasping the opportunities of a strategy based on the ecological footprint. In this case, firms' expectations of such strategies showed effects of virtuous behaviours (i.e., experience-based resources), whose rents have been capitalised and captured mainly by improvements of image and reputation and customers' relationships loyalty, entrance into new foreign markets. These outcomes seem to exploit a certain degree of perceived use value increases (i.e., through superior sales volumes and even premium prices).

Despite the attempt to give the best possible picture of Italian firms' CFS, our study has some limitations. First, although the firms' sample includes one-fifth of all the Italian wine producers who have adopted a CF certification its size is relatively small. This fact implies that the findings of the study should be interpreted with caution. Second, the exploratory nature of this research does not handle with any specific set of hypotheses, although scientific progress often stems from observations, made by chance or by project. Third, a part of the survey data are related to opinions, perceptions, and expectations expressed by the firms representatives, making them affected by cultural and regional-specific characteristics that may have an impact on manager responses. Fourth, the study topic on the carbon footprint as a lever for sustained competitive advantages is marked by its complexity. Therefore, in attempting to provide a first portrayal of the state-of-the-art on CFS in the Italian wine industry some simplifications were made. In trying to overcome the limitations of the study, solutions were sought by limiting the established sophistication of the study to firms' core processes within the supply chain management (i.e., core resources deployed and strategies).

More interestingly, by looking at some research limitations, tentative paths for further analysis within this topic originate [91]. In order to confirm our findings, new studies in other world wine-growing areas may be useful. Moreover, the adoption of other macro indicators of sustainability, which in other words, could follow the way of operation of a complex phenomenon (i.e., like a kind of "isolating mechanism" as argued by Rumelt [46], are suggested in order to explore further sources of sustained competitive advantage [92].

To conclude, in a climate change scenario, it might be expected that valuing firms' sustainable marketing through activities aimed at improving CF performance can be a win-win strategy [35,54,93] for its implications on both firms competitiveness and environmental goals.

**Supplementary Materials:** The following are available online at http://www.mdpi.com/2071-1050/11/5/1483/s1: The dataset, Figure 1 and questionnaire.

**Author Contributions:** Conceiving Research and Designing Research Framework, L.G. and L.B.; Collecting and Analyzing Data, L.G. and L.B.; all authors wrote and reviewed the paper.

**Funding:** This research received no external funding.

**Acknowledgments:** The authors would like to thank Marco Tonni (Agronomi Sata), Stefano Stefanucci (Equalitas), Maria Dei Svaldi (V.I.V.A.—Sustainable Wine) for their valuable helps and assistance with the collection of data in Section 3 (Table A1).

**Conflicts of Interest:** The authors declare no conflicts of interest.

# Appendix A

**Table A1.** Italian firms with carbon footprint certifications, 2018.

| Regions of Italy | Supply Chain | Wine Production | Wines Produced | Firm Carbon Footprint | | | Data Source |
|---|---|---|---|---|---|---|---|
| | *Phases* | *Bottles* | *Types* | *Grapes production* | *Processing into wine* | *Bottling and packaging* | |
| | | | | $Kg$-$CO_2$ eq per 1 ton | $Kg$-$CO_2$ eq per 1 L | $Kg$-$CO_2$ eq per 1 bottle (0.75 L) | |
| **North-West** | Grapes, bulk and bottling wine production | 97,171 | White, red/rosè, sparkling | 4.8346 | 0.0400 | 0.8080 | *Equalitas* |
| **Central** | Grapes, bulk and bottling wine production | 293,861 | White, red/rosè | 0.3675 | 0.2130 | 0.5860 | *Equalitas* |
| **Central** | Grapes, bulk and bottling wine production | 1,045,915 | White, red/rosè | 5.8455 | 0.4590 | 0.4370 | *Equalitas* |
| **Central** | Grapes, bulk and bottling wine production | 1,444,489 | White, red/rosè | 1.1654 | 0.2300 | 0.2310 | *Equalitas* |
| **Central** | Grapes, bulk and bottling wine production | 55,759 | White, red/rosè | 2.1928 | 0.3785 | 0.5250 | *Equalitas* |
| **Central** | Grapes, bulk and bottling wine production | 201,254 | White, red/rosè, sparkling | n/a | 0.1923 | 0.8510 | *Equalitas* |
| **Central** | Grapes, bulk and bottling wine production | 3,583,600 | White, red/rosè, sparkling | n/a | 0.1892 | 0.4310 | *Equalitas* |
| **South and Islands** | Grapes, bulk and bottling wine production | 2,500,000 | White, red/rosè, sparkling | n/a | n/a | n/a | *Equalitas* |
| **South and Islands** | Bulk and bottling wine production | 700,000 | White, red/rosè, sparkling | n/a | n/a | n/a | *Equalitas* |
| | *Phases* | *Bottles* | *Types* | *Ambit 1* | *Ambit 2* | *Ambit 3* | |
| | | | | t $CO_2$ eq | | | |
| **North-West** | Grapes, bulk and bottling wine production | 275,000 | White, red/rosè | $1.37 \times 10^2$ | $6.15 \times 10^1$ | $2.40 \times 10^2$ | *VIVA* |
| **North-West** | Grapes, bulk and bottling wine production | 500,000 | White, red/rosè | $1.50 \times 10^5$ | $5.37 \times 10^5$ | $7.41 \times 105$ | *VIVA* |
| **Central** | Grapes, bulk and bottling wine production | 208,000 | White, red/rosè | $4.69 \times 10^1$ | $3.76 \times 10^{-3}$ | $1.26 \times 10^2$ | *VIVA* |
| **South and Islands** | Grapes, bulk and bottling wine production | 2,300,000 | White, red/rosè | $7.23 \times 10^2$ | $4.00 \times 10^2$ | $2.19 \times 10^3$ | *VIVA* |
| **South and Islands** | Grapes, bulk and bottling wine production | 3,000,000 | White, red/rosè | $3.50 \times 10^2$ | $3.37 \times 10^2$ | $4.76 \times 10^3$ | *VIVA* |

Source: Authors' processing on Agronomi Sata, Equalitas and VIVA data, 2018.

**Table A2.** Descriptive statistics of structural and market data, 2017.

| Variables | Percentage | Mean | St. Dev | Min | Max |
|---|---|---|---|---|---|
| **Vineyard surface** *(hectares)* | | 106.062 | 84.162 | 7 | 262.42 |
| **Cellar surface** *($m^2$)* | | 3636.500 | 3706.165 | 200 | 13,200 |
| **Turnover** *(4 categories):* | | | | | |
| **-Below 1,000,000 €** | 30% | | | | |
| **-From 1,000,000 to 5,000,000 €** | 40% | | | | |
| **-From 5,000,000 to 10,000,000 €** | 20% | | | | |
| **-Over 10,000,000 €** | 10% | | | | |
| **Environmental certifications** *(number)* | | 1.4 | 0.699 | 1 | 3 |
| **Vineyard employees** *(number)* | | 8.0 | 5.354 | 1 | 18 |
| **Cellar employees** *(number)* | | 6.3 | 4.715 | 1 | 16 |
| **Adm. and marketing employees** *(number)* | | 6.7 | 6.290 | 0 | 23 |
| **Total sales** *(bottles):* | | 1,288,287 | 1,578,040 | 15,000 | 4,520,000 |
| **Red and rosè wines** *(bottles sold)* | | 481,316 | 743,680 | 6000 | 2,500,000 |
| **White wines** *(bottles sold)* | | 726,286 | 103,464 | 1000 | 3,000,000 |
| **Sparkling wines** *(bottles sold)* | | 70,638 | 180,215 | 0 | 580,967 |
| **Market share of export markets** *(%)* | | 43.6% | 26.7% | 5.0% | 90.0% |

Source: author's elaboration.

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
