# Peer review of "Carbon Footprint as a Lever for Sustained Competitive Strategy in Developing a Smart Oenology: Evidence from an Exploratory Study in Italy"

_sustainability, doi:10.3390/su11051483_

Round 1

Reviewer 1 Report

The subject of the article is interesting and it is, somehow, linked to the objectives of the journal (a dopey explanation of the relation with sustainability is, anyway, desirable). However, there are a number of issues that have to be reconsidered.

For a better visibility on databases, the authors are asked not to repeat among keyword the words/concepts included on the title of the article 

Lines 201-202. What “some firms” mean? The additional explanation on how they were selected, why they are representative for the entire studied population is to be done.

Lines 210-217. The pretesting process has to be deeply descripted. When? Where? How? Etc.

Lines 268-269. There is no clear how the export countries/regions were defined. Additional information could help understanding this share.

Lines 300-307. There are necessary more explanation why the sample is representative, the approaches of the authors is too superficial at the moment.

A clear separation between discussions and results can help readers to understand the article.

Author Response

Answers to Reviewer 1

General improvements:

Description of methods was improved with reference to the firms sample selection and its representativeness (Lines 312-323) and the pretesting process is now more detailed (Lines 211-225).

Moreover, the Discussion (lines 513-565) is now separated from the Conclusion (lines 568-636). The links between Results, Discussion and Conclusions are now more clearly established.

Specific topics:

Keywords now do not include words that are in the title and 4 l new significant words are added.

A further explanation on how the sample firms were selected are selected and why the can be considered representative for the entire studied population is given in lines 312-323. The words “some firms” in line 201 do not refer to firms included in the sample, but to wineries more or less focused on sustainability that were broadly overview in order to better define the questions to be included in the questionnaire.

A more accurate definition of export countries/regions were defined in lines 276-282.

Thank you very much for the valuable review.

Reviewer 2 Report

The paper " Carbon Footprint as a Lever for Sustained Competitive Strategy in Developing a Smart Oenology: Evidence from An Exploratory Study in Italy" is of interest for the readers of Sustainability, and analyzes a relevant topic: sustainability as a marketing strategy.

The article seems a bit long. Maybe some of the text could be provided as supplementary materials. Several comments to further improve it:

- Abstract is a bit longer than recommended (200 words)

- Use 2019 template

- Keywords could be improved

- Are EPD available for wine?

- 2. Conceptual framework

- Figure 1: Where are Winery J and K?

- Explain the possible relationships between photovoltaic energy and Italian goverment aids.

- Clearly explain the limitations of the study

- The link between results and "concluding discussion" could be clearly established.

Author Response

Answers to Reviewer 2

General improvements:

In order to improve the readability of the paper the Discussion is now separated from the Conclusion and it is more linked to the main outcomes illustrated in the previous paragraph.

Given the difficulties faced in shortening the article, we asked the Editor about the article length and we have the assurance that it its length is compatible with the Journal standards. The questionnaire and the dataset are already available as annexed materials.

Specific topics.

Abstract was a bit shortened. Now it accounts for 178 words.

2019 template is now adopted.

Keywords are improved by additional words.

EPD are available for wine. However, they were not included in the questionnaire because, when we gathered preliminary information in order to structure the questionnaire, they appeared not to be of any interest of the investigated wineries.

In Figure 1 Wineries J and K were not present because we listed the wineries according to the Italian alphabet where J and K are not included. The new figure 1 eliminates this mistake.

The relationships between photovoltaic energy and government aids is now considered in the sixth paragraph (lines 523-528).

Limitations of the study are now better clarified (lines 514-564).

Discussion has been deeply revised with additional comments on the main results (lines 600-614).

Thank you very much for the valuable review.

Round 2

Reviewer 1 Report

The article was improved by authors.